# Polymerase-guided base editing enables in vivo mutagenesis and rapid protein engineering

Aaron Cravens[1,5], Osman K. Jamil[2,5], Deze Kong[1], Jonathan T. Sockolosky[3] & Christina D. Smolke [1,4]✉

Random mutagenesis is a technique used to generate diversity and engineer biological systems. In vivo random mutagenesis generates diversity directly in a host organism, enabling applications such as lineage tracing, continuous evolution, and protein engineering. Here we describe TRIDENT (TaRgeted In vivo Diversification ENabled by T7 RNAP), a platform for targeted, continual, and inducible diversification at genes of interest at mutation rates one-million fold higher than natural genomic error rates. TRIDENT targets mutagenic enzymes to precise genetic loci by fusion to T7 RNA polymerase, resulting in mutation windows following a mutation targeting T7 promoter. Mutational diversity is tuned by DNA repair factors localized to sites of deaminase-driven mutation, enabling sustained mutation of all four DNA nucleotides at rates greater than $10^{-4}$ mutations per bp. We show TRIDENT can be applied to routine in vivo mutagenesis applications by evolving a red-shifted fluorescent protein and drug-resistant mutants of an essential enzyme.

[1] Department of Bioengineering, 443 Via Ortega, MC 4245, Stanford University, Stanford, CA, USA. [2] Department of Chemical Engineering, 443 Via Ortega, MC 4245, Stanford University, Stanford, CA, USA. [3] Departments of Molecular and Cellular Physiology and Structural Biology, Stanford University School of Medicine, Stanford, CA, USA. [4] Chan Zuckerberg Biohub, San Francisco, CA, USA. [5]These authors contributed equally: Aaron Cravens, Osman K. Jamil. ✉email: csmolke@stanford.edu

Generation of genetic variation through random mutagenesis is fundamental to the study and engineering of biology in applications as diverse as protein engineering, directed and continuous evolution, and lineage tracing[1–3]. Conventional random mutagenesis requires manipulation of DNA ex vivo and transformation of DNA libraries into host cells, requires cellular hosts with high transformation efficiencies limiting library size to the number of transformants, and is incompatible with continuous evolution or lineage tracing. Targeted in vivo random mutagenesis can overcome these limitations by continuously generating diversity in vivo via localization of mutagenic enzymes or nucleases to DNA[1].

In the past decade, advances have been made in engineering systems for targeted in vivo mutagenesis; however, current in vivo mutagenesis techniques are limited by low rates of DNA mutagenesis, inability to tune the spectrum of mutations, restrictive requirements such as the use of specific host/viral pairings[2], orthogonal replication systems[3,4], or the need to multiplex dozens of guide-RNAs (gRNA) to tile a target gene[1]. For example, recent platforms for targeted in vivo mutagenesis have relied on Cas9-variants fused to mutagenic polymerase or deaminase enzymes[5]. In CRISPR-X, expression of a gRNA targeted to a specific gene results in deaminase-induced mutation adjacent to the target site[6] and in EvolvR, expression of an error-prone nick-translating DNA polymerase fused to enCas9 yields mutation rates of $10^{-2}$ substitutions per base (s.p.b.) one base pair from the enCas9 induced nick[1,7]. For these CRISPR-based targeting systems, mutations are limited to an approximately 40 bp window adjacent to the gRNA target site, mutation of the 20 bp gRNA binding site is inhibited, and mutagenesis is focused to just two or three base pairs. For example, the mutation rate drops from $10^{-2}$ at 1 bp to $10^{-6}$ at 56 bp from the Cas9 nick site in the EvolvR system. Thus, effective diversification of a 1000 bp gene can require more than 20 gRNAs, requiring redesign for each new target and inhibiting mutation at over 400 bp of sequence corresponding to gRNA binding sites. Further, the single mechanism of mutation (e.g., deaminase or polymerase) results in a fixed mutation bias which cannot be adjusted to the needs of the user[6]. These constraints have limited application of in vivo mutagenesis to making targeted point mutations or to applications compatible with low mutation rates such as evolving growth-linked antibiotic tolerance. During the preparation of this manuscript, several polymerase-targeted mutagenesis systems were reported that address some of these limitations[8–10]. For example, TRACE targets a mutagenic cytidine deaminase enzyme and uracil DNA glycosylase inhibitor protein (Ugi) to DNA by fusion to T7 RNA Polymerase (T7 RNAP)[10]. While TRACE enables continuous mutagenesis in human cells across multiple kilobases of DNA, it lacks chemically inducible tunability of overall mutation rate and has low diversity of mutations, with a notable bias to C/G transition mutations. Moore et al. and Álvarez et al. also developed a cytidine deaminase T7 RNA Polymerase targeting system in E. coli, but did not utilize DNA-modifiers to increase mutation diversity and both systems, while more targeted than chemical mutagenesis, still retained significant off-target mutation frequency[8,9].

Here, we developed a method for TaRgeted In vivo Diversification ENabled by T7 RNAP (TRIDENT). TRIDENT is a platform that achieves in vivo mutagenesis rates exceeding $10^{-3}$ s.p. b., >1000-fold on-target mutation specificity, and mutational diversity at all four DNA nucleotides which can be tuned over a 100-fold range by the user. TRIDENT consists of up to three components targeted to a locus of interest via T7 RNA polymerase: (1) a cytosine deaminase enzyme that introduces C>T and G > A substitutions, (2) an engineered adenosine deaminase enzyme that introduces A > G and T > C substitutions, and (3)

DNA-repair factors that further enhance A/T/G mutational diversity.

The core of TRIDENT is the targeted recruitment to DNA of synthetic fusion proteins comprising T7 RNAP and mutagenic enzymes, leveraging the processive action of the polymerase to target mutations to DNA. TRIDENT takes inspiration from the highly targeted mutagenesis that occurs in B-cell somatic hypermutation (SHM), in which the cytosine deaminase enzyme AID is a key initiator. SHM co-transcriptionally targets AID to ssDNA at the immunoglobulin locus[11,12], where it deaminates cytosine to uracil resulting in accumulation of U:G mismatches. Mismatches are subsequently processed by replicative polymerases yielding C>T mutations, or error-prone forms of base excision repair (BER) and mismatch repair (MMR) to yield transversions and transitions at A, T, C, and G and an overall targeted mutation rate of $10^{-3}$ s.p.b., $10^5$–$10^6$-fold higher than genomic error rates[13].

## Results

**Targeted in vivo mutagenesis with a cytidine deaminase–polymerase fusion.** We first demonstrated that T7 RNAP fused to the hyperactive AID homolog PmCDA1[14] can be recruited to target DNA, defined by a T7 promoter sequence, where it increases mutation rate (Fig. 1a). The mutagenic activity of the PmCDA1-T7 RNAP fusion was characterized in a yeast strain (YHM0) that harbors an expression cassette for the selection marker URA3 with an upstream T7 promoter ($P_{T7}$) integrated into the genome to recruit the T7 RNAP fusion and a disruption in the UNG1 gene, encoding the primary enzyme responsible for DNA repair of uracil, to increase mutation rate[15] (Supplementary Fig. 1). We designed a fluctuation assay to measure the occurrence of loss-of-function (LOF) mutations at URA3 by plating cells on 5-FOA plates, which inhibit growth of $URA3^+$ cells. URA3 LOF colony counts are converted to URA3 LOF mutation rate using the Falcor algorithm[16]. We performed the assay on strains with and without $P_{T7}$ upstream of URA3 (YHM0 and YHM0.$P_{T7}$KO, respectively) expressing either PmCDA1 (pCS4312) or PmCDA1-T7 RNAP (pCS4314) induced with galactose for 16–24 h. The data show elevated mutation rates only in the presence of both the PmCDA1-T7 RNAP fusion and $P_{T7}$ (Fig. 1b). Targeted mutation by PmCDA1-T7 RNAP resulted in mutation rates of up to $1.1 \times 10^{-2}$ LOF mutations per division, a 690,000-fold increase from the wild-type background LOF rate of $1.6 \times 10^{-8}$ (Supplementary Fig. 1).

We next optimized the platform to improve overall quality of targeting by reducing off-target genomic mutations while maintaining a high on-target mutation rate. The off-target mutation rate is measured via the fluctuation assay in a strain lacking $P_{T7}$ upstream of URA3 (YHM0.$P_{T7}$KO, Supplementary Fig. 2). Plasmid-based expression of PmCDA1-T7 RNAP increased off-target mutagenesis by more than 2500-fold above the background rate, resulting in a net on:off-target ratio of 60-fold (Fig. 1b). We tested five strategies to improve the target ratio, including optimizing the linker peptide length between PmCDA1 and T7 RNAP and adding DNA binding domains to PmCDA1-T7 RNAP in order to enhance localization to $P_{T7}$ (Supplementary Fig. 3). The combined strategy of controlling the expression of PmCDA1-T7 RNAP from a synthetic, inducible promoter[17] and integrating the expression cassette for the fusion protein into the genome of the base strain YHM0 (resulting in YHM1) resulted in the highest on:off-target ratio of up to 1800-fold when induced with 2 nM β-estradiol inducer for 16 h (Fig. 1c). The improved system further enabled titratable control of on-target mutation rate over a 100-fold range from a LOF rate of $2.5 \times 10^{-5}$ to $2.7 \times 10^{-3}$ (Fig. 1c).

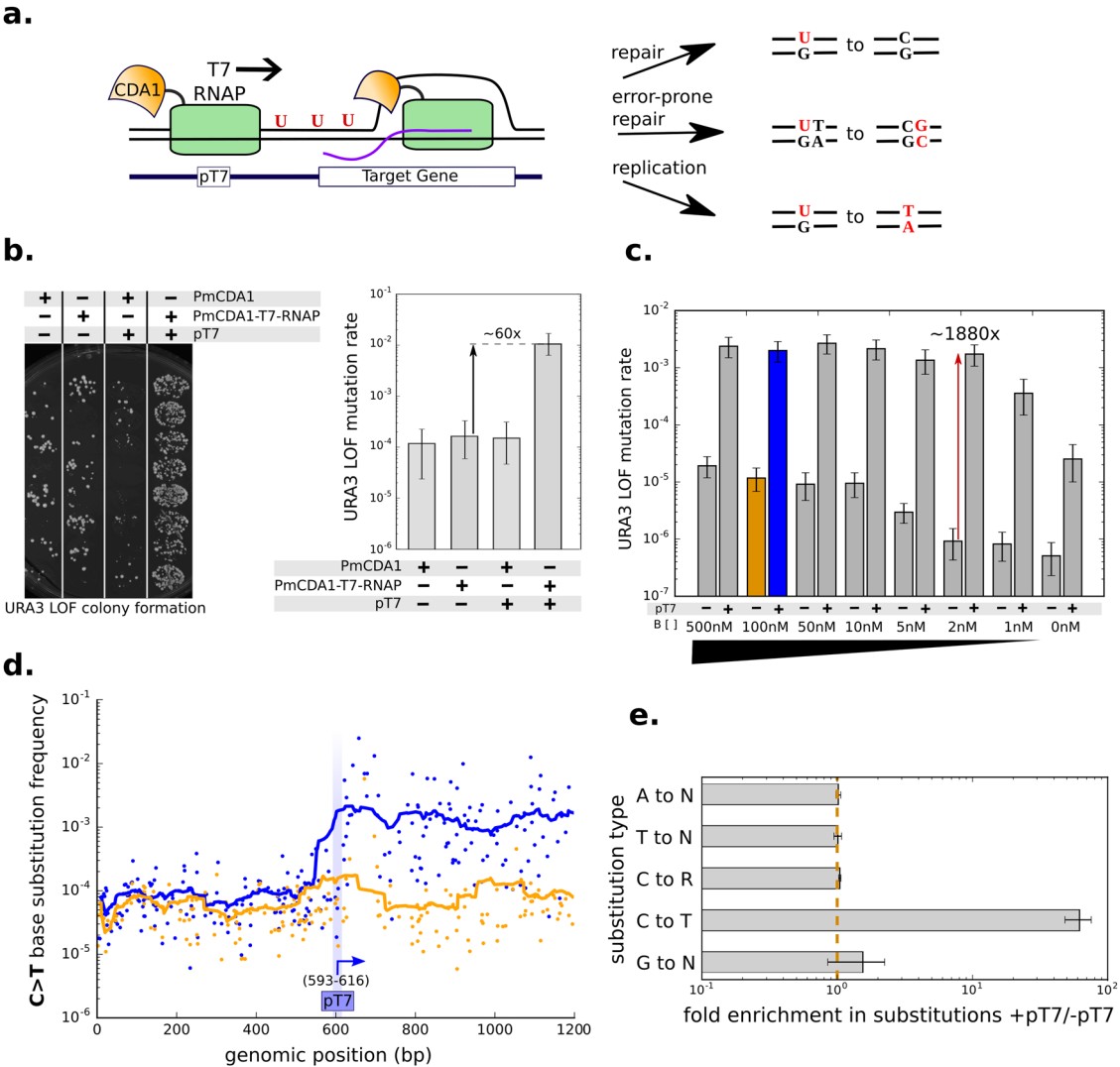

**Fig. 1 A cytosine deaminase T7 RNA polymerase fusion allows for in vivo targeting of mutagenesis. a** Schematic of mutagenesis targeting system and types of mutations that are produced as a result of U:G mismatches resulting from action of PmCDA1 on cytosine. ssDNA substrate is made available to PmCDA1 as a result of T7 RNAP activity (RNA in purple). In this work the target gene is *URA3*, and $P_{T7}$ is placed upstream of $P_{URA3}$. **b** Mutation rate at *URA3* visualized by *URA3* loss-of-function (LOF) colony growth on 5-FOA (left) and quantified using the Falcor algorithm (right). PmCDA1 is expressed alone (pCS4312) and as a fusion to T7 RNAP (pCS4314) in strains YHM0 and YHM0.$P_{T7}$KO. Error bars represent 95% confidence intervals measured from eight independent cultures induced with galactose. **c** Mutation rate at *URA3* quantified by Falcor algorithm for strains with PmCDA1-T7 RNAP integrated under control of an inducible promoter. On- (YHM1) and off-target (YHM1.$P_{T7}$KO) mutation rates are quantified for varying concentrations of inducer. Error bars represent 95% confidence intervals measured from 8 independent cultures induced with specified concentration of β-estradiol. **d** Mutation frequencies for C>T mutations plotted by position along the *URA3* sequence for YHM1 (blue) and YHM1.$P_{T7}$KO (yellow) when induced with 100 nM β-estradiol. In order to match YHM1 and YHM1.$P_{T7}$KO, data from YHM1.$P_{T7}$KO is off-set to account for the gap created by the 23 bp insertion which disrupts $P_{T7}$. Mutation data obtained by next-generation sequencing (NGS) at the *URA3* locus as described in "Methods". **e** The ratio of substitutions per base (s.p.b.) between strains YHM1 and YHM1.$P_{T7}$KO indicates change in mutation rate resulting from addition of $P_{T7}$ targeting. The average s.p.b. was calculated for each substitution type between bases 800 and 1000 bp using the same NGS data from (**d**). Fold enrichment in substitutions was calculated by dividing the average s.p.b. in the presence of $P_{T7}$ by the average s.p.b. in the absence of $P_{T7}$, and error bars represent standard deviation calculated between two biological replicates, each with an average computed from $N = 200$ bps. Source data are available as a Source data file.

TRIDENT with PmCDA1 achieves several important characteristics of engineered in vivo mutagenesis, including precise initiation of mutation and constant sustained mutation rate across a target region. We induced mutagenesis in YHM1 with 100 nM β-estradiol for 16 h and performed targeted deep sequencing at the $P_{T7}URA3$ locus on extracted genomic DNA, spanning a region up to 2 kb downstream of $P_{T7}$. The data demonstrate that initiation of C > T mutagenesis is precise, occurring immediately following $P_{T7}$ with the first increase in mutation rate occurring at cytosines that are 9–12 bases from the

RNA transcription initiation site of T7 RNAP (Fig. 1d, Supplementary Fig. 4).

The results further demonstrate that TRIDENT can target mutagenesis over multiple kilobases of sequence (Supplementary Fig. 5). Mutation of the $P_{T7}$ sequence was minimal, with no single base substitution frequency within $P_{T7}$ >$2 \times 10^{-4}$, indicating that <0.1% of cells lose targeting activity during the 16 h of mutation induction (Supplementary Fig. 4). However, in analyzing the types of mutations introduced at the target locus by calculating the ratio of mutant alleles in the presence and absence of $P_{T7}$, the

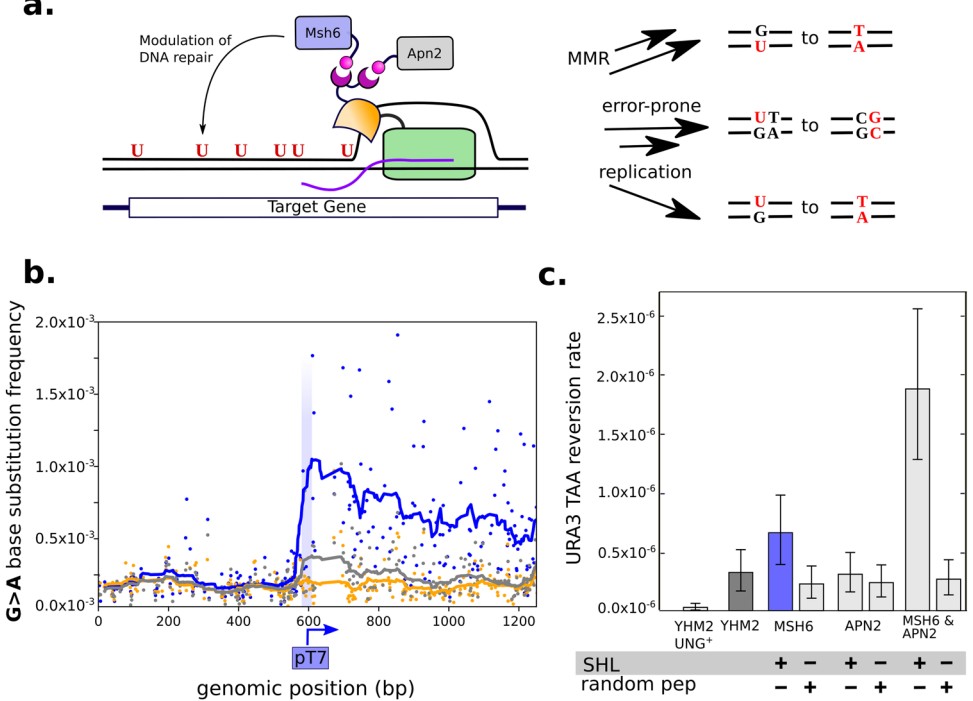

**Fig. 2 Mutational spectrum is tuned by engineered targeting of repair factors. a** Schematic of DNA-repair factor targeting using SH3 domains (purple) fused to integrated PmCDA1-T7 RNAP and SHL domains (pink) fused to host DNA-repair proteins. Localization of repair proteins initiates different mutational outcomes compared to PmCDA1-T7 RNAP alone. **b** Mutation frequencies for G > A substitutions plotted by position along *URA3* for YHM2. MSH6 (blue), YHM2 (gray), and YHM2.P$_{T7}$KO (yellow) when induced with 100 nM β-estradiol. In order to match YHM2 and YHM2.P$_{T7}$KO, data from YHM2.P$_{T7}$KO is off-set to account for the gap created by the 23 bp insertion which disrupts P$_{T7}$. Mutation frequencies for all other base substitutions are shown in Supplementary Fig. 8b. Mutation data generated by next-generation sequencing at the *URA3* locus as described in "Methods". **c** A/T targeted mutation rate in strain YHM2.TAA having Apn2p and Msh6p DNA-repair proteins tagged with an SHL peptide or short random peptide. A/T rate is measured by reversion of a TAA stop codon in *URA3* and quantified using the Falcor algorithm. Error bars represent 95% confidence intervals measured from 24 independent cultures induced with 100 nM β-estradiol. Source data are available as a Source data file.

data indicate that over 99% of induced mutations are C > T transitions (Fig. 1e, Supplementary Fig. 6).

**Localization of DNA-repair enzymes alters mutation spectrum**. We expanded the TRIDENT platform to target DNA-repair factors involved in A/T/G mutagenesis in SHM to enhance mutational diversity. AID-catalyzed deamination of cytosines yields U:G mismatches, which are subsequently processed by error-prone DNA-repair pathways[18] to yield equal diversity across all bases during SHM. We co-localized yeast homologs of twelve DNA-repair factors involved in SHM A/T/G mutagenesis to the T7 RNAP fusion using modular SHL/SH3 domains[19] (Fig. 2a, Supplementary Table 2). PmCDA1-T7 RNAP was modified with an N-terminal SH3-SH3 peptide tag (YHM2) to localize proteins harboring a SHL peptide to T7 RNAP (Supplementary Fig. 3c). We used genome editing[20] to fuse SHL tags to the N- or C-termini of each endogenous DNA-repair protein. A yeast strain was engineered with a premature stop codon (TAA) at the Q8 residue of *URA3* (YHM2.TAA), such that the fluctuation assay could be used to measure gain-of-function (GOF) mutations occurring at the TAA via non-C mutations. Results of the fluctuation assay performed in YHM2.TAA with the 12 candidate SHL-fusions show that the highest rate of A/T mutagenesis occurred with SHL targeting of the mismatch repair protein Msh6p (YHM3.TAA, Supplementary Fig. 7). Sequencing of the P$_{T7}$*URA3* loci from 16 individual YHM3.TAA colonies indicated that all GOF mutations were T > G transversions at TAA, suggesting that error-prone DNA-repair mechanisms generate mutations distinct from those of the cytosine deaminase.

Deep sequencing results further revealed a 5-fold enrichment of G > A substitutions in YHM3.TAA (average G > A mutation rate of $5 \times 10^{-4}$ s.p.b.) compared to the non-SHL-targeted strain YHM2 (Supplementary Fig. 8), though the deep sequencing data suggest the G > A substitution rate may drop-off with distance from the T7 promoter (Fig. 2b). The observed increase in G > A substitutions likely results from increased Msh6p-associated MMR at *URA3*, resulting in excision of the coding strand to expose the non-coding strand for deamination by PmCDA1. In support of this model, SHL-tagging of the MMR-associated proteins Exo1p and Rad18p also resulted in increased G > A mutagenesis (Supplementary Fig. 8).

We next demonstrated synergistic effects of repair factor targeting on the rate of A/T mutagenesis by co-localizing two DNA-repair factors with T7 RNAP. We constructed sixteen new yeast strains with combinatorial pairs of SHL tags fused to endogenous DNA-repair proteins (Supplementary Fig. 9), measured the TAA-reversion rate for each in the GOF fluctuation assay, and selected a subset with high TAA-reversion rates for further validation. In particular, the co-localization of SHL-Msh6p and SHL-Apn2p to PmCDA1-T7 RNAP (YHM4) resulted in a 5-fold higher TAA-reversion rate than YHM2 and exhibited an overall TAA codon mutation rate of $2 \times 10^{-6}$ (Fig. 2c). We showed that localization of both DNA-repair proteins to the T7 RNAP fusion was necessary for the observed increase in A/T mutation rate, as identical strains in which the SHL tag was replaced by a short non-functional random peptide sequence (YHM2.TAA.pep-Msh6p, YHM2.TAA.pep-Apn2p, and YHM2. TAA.pep-Apn2p.pep-Msh6p) do not display higher TAA-reversion rates. Deep sequencing of the P$_{T7}$*URA3* locus in

YHM4 indicated that C > T s.p.b. remained comparable to parent strain YHM2 despite the modifications to repair processing (Supplementary Fig. 10). Taken together, the data show that SHL targeting of DNA-repair factors increased rates of A/T mutation by up to 5-fold and increased rates of G mutagenesis by at least 5-fold. However, the overall mutation spectrum remained biased as over 99% of induced mutations are C > T and G > A substitutions (Supplementary Fig. 11).

**An engineered adenosine deaminase increases A/T mutations and acts synergistically with SHL-targeted repair factors to increase mutation diversity**. Finally, in order to balance the rates of A/T and C/G mutagenesis, we expanded the TRIDENT platform to target an adenosine deaminase, which deaminates adenosine to inosine resulting in A-to-I-to-G transitions. We started with a previously reported engineered adenosine deaminase derived from base editor BE7.10, consisting of a heterodimer of the *Escherichia coli* tRNA adenosine deaminase enzyme (TadA) fused to an engineered variant (TadA*) containing 14 mutations, further fused to dCas9[21]. We created a fusion of the engineered TadA-TadA* heterodimer and T7 RNAP (TadA-TadA*-T7 RNAP, pCS4332). The activity of TadA-TadA*-T7 RNAP was measured in YHM1.TAA as the TAA-reversion rate in a GOF fluctuation assay. The data show that expression of TadA-TadA*-T7 RNAP resulted in no significant change in the targeted TAA-reversion rate (Fig. 3b). As TadA-TadA* was engineered in the context of dCas9 targeting, we hypothesized that the new protein context as a fusion to T7 RNAP resulted in reduced adenosine deaminase activity.

We used directed evolution and selection to engineer an adenosine deaminase enzyme that exhibited increased A/T targeted mutations when fused to T7 RNAP (Fig. 3a). We built a selection strain containing a TAA stop codon in P$_{T7}$*URA3* (CSY1256) to enrich for TadA-TadA*-T7 RNAP variants with increased TAA-reversion rates, resulting in GOF growth on media lacking uracil. A library of TadA-TadA* mutants fused to T7 RNAP was generated via PCR mutagenesis and transformed via gap repair with pCS3414 into the CSY1256 selection strain. Expression of the TadA-Tad*-T7 RNAP variants was induced with galactose for 16 h and cells were subsequently plated on media lacking uracil to select for A/T mutations in the TAA stop codon of *URA3*. Yeast colonies that grew were pooled and the plasmids extracted and utilized in the next round of library mutagenesis and selection. We observed a substantial increase in the number of TAA reverting colonies on media lacking uracil after five rounds of evolution (Supplementary Fig. 12). The TadA-Tad*-T7 RNAP variants were sequenced from 16 colonies to determine consensus mutations and we found that in all colonies the TadA-TadA* heterodimer was reduced to a TadA* monomer containing 4 new consensus mutations (this yeast evolved variant was termed yeTadA1.0, Fig. 3b). In the GOF fluctuation assay in YHM1.TAA the yeTadA1.0-T7 RNAP fusion (pCS4333) exhibited a 270-fold increase in TAA-reversion rate compared with the TadA-TadA*-T7 RNAP fusion (pCS4332). In addition, the yeTadA1.0-T7 RNAP fusion exhibited a 1570-fold on:off-target ratio based on measured TAA-reversion rates in YHM1.TAA and YHM1.TAA.P$_{T7}$KO.

Expansion of the TRIDENT platform by incorporation of the engineered yeTadA1.0-T7 RNAP fusion enables precise, targeted mutagenesis at A/T and functions alongside C/G mutagenesis generated with PmCDA1 and SHL targeting. We integrated an expression cassette encoding yeTadA1.0-T7 RNAP under control of a β-estradiol inducible promoter[17] into the genome of strains YHM2 and YHM4 to create strains YHM5 and YHM6, respectively. We induced mutagenesis in YHM5 and YHM6 with

100 nM β-estradiol for 16 h and performed targeted deep sequencing at the P$_{T7}$*URA3* loci on extracted genomic DNA. The results demonstrate that YHM5 and YHM6 exhibited enhanced A/T mutagenesis compared to YHM2 and YHM4 as a result of the localization of the engineered yeTadA1.0 deaminase by T7 RNAP (Fig. 3c). Initiation of A/T mutagenesis in YHM5 and YHM6 is constrained to the region downstream of P$_{T7}$ and the mutation frequency within the P$_{T7}$ sequence remained below $5 \times 10^{-4}$ at any single base position when induced with 100 nM β-estradiol. The data also show that G:A mutation frequencies were increased in YHM6 compared with YHM5 and YHM2, indicating that SHL-Msh6p and SHL-Apn2p enhance base diversity even in the presence of the engineered yeTadA1.0 (Fig. 3c). We also confirmed that the most advanced TRIDENT strains, YHM5 and YHM6, retained similar on and off-target mutation rates to YHM2, which had been optimized for lower off-target mutagenesis (Supplementary Fig. 13). Finally, we showed that the TRIDENT system has minimal impact on cellular fitness by measuring the growth rates of strains YHM2, YHM5, and YHM6 induced with β-estradiol. There was no statistically significant difference ($p < 0.05$) in growth rate for any level of β-estradiol induction up to 100 nM (Supplementary Table 3).

The most advanced TRIDENT strain, YHM6, features in total seven genomic modifications to achieve greater than $10^{-4}$ s.p.b. across all four nucleotides (Fig. 3d, Supplementary Fig. 14). A/T mutation is generated primarily via T7 RNAP targeting of an engineered adenosine deaminase enzyme that introduces A > G and T > C substitutions and secondarily through SHL targeting of Apn2p and Msh6p. C mutagenesis is generated via T7 RNAP targeting of PmCDA1, and G mutagenesis is generated primarily via SHL targeting of Msh6p. Importantly, while in this work we implemented β-estradiol inducible control of both PmCDA1 and yeTadA1.0, independent control of A/T and C/G bias could be implemented by using orthogonal inducible promoters. Additional modifications include integration of P$_{T7}$*URA3* to enable mutation rate assessment, integration of a ZEV synthetic transcription factor to support β-estradiol inducible promoter usage[17], and disruption of the *UNG1* gene to improve C/G mutagenesis.

**TRIDENT enables rapid protein engineering**. The most advanced TRIDENT platform (YHM6) was demonstrated to enable rapid in vivo protein evolution for three protein engineering challenges (Figs. 4 and 5). Recent state-of-the art in vivo mutagenesis systems have been applied to engineer proteins using either screens or selections. Most protein activity cannot be linked to growth and is therefore screened through lower throughput assays, such as mass spectrometry or fluorescence-activated cell sorting (FACS). In contrast, growth limiting genes, such as antibiotic resistance markers, can be evolved using a growth rate or survival-based selection[1]. We demonstrated the flexibility of the TRIDENT platform by applying it to protein evolution challenges that rely on both screens and selections.

The TRIDENT platform is capable of evolving non-selectable traits using FACS and able to make single base substitutions at a known position in mCherry. We integrated an expression cassette encoding mCherry containing a premature TAA stop codon at N13 flanked by a T7 promoter 600 bases upstream of mCherry into the genome of YHM6 (YHM6.mCherry.TAA). We induced mutagenesis in YHM6.mCherry.TAA with 10 nM β-estradiol for 16 h and sorted for mCherry-positive cells by FACS, observing a 0.12% mCherry-positive population (Supplementary Fig. 15). The mCherry-positive population was reanalyzed to confirm mCherry expression (Fig. 4b) and individual colonies were sequenced to

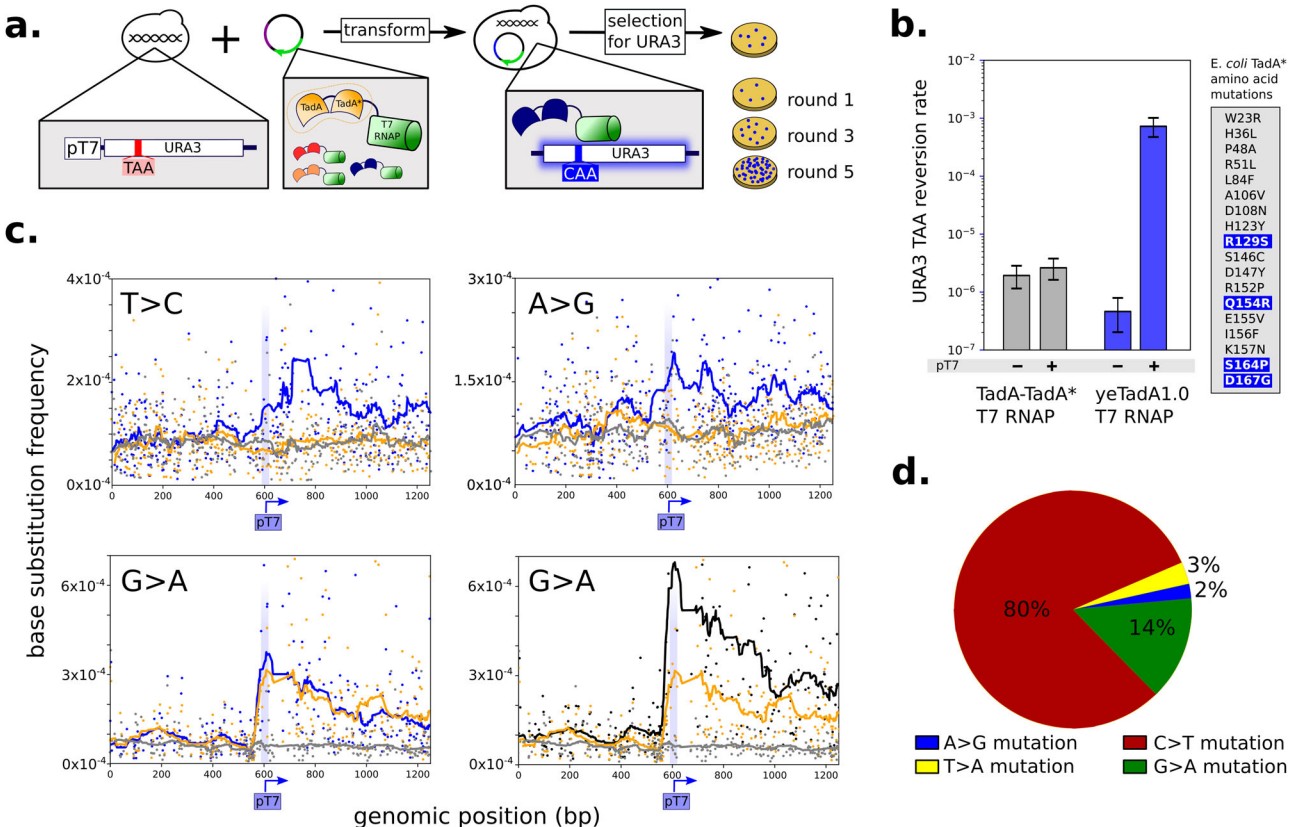

**Fig. 3 An engineered adenosine deaminase increases targeted A/T mutation rate. a** Schematic of strategy to evolve an adenosine deaminase for targeted mutagenesis when fused to T7 RNAP. Screening strain containing *URA3* with a premature TAA stop codon is transformed with libraries of TadA-TadA* adenosine deaminase fused to T7 RNAP (left). TadA-TadA* variants with enhanced A/T mutagenesis correct the TAA stop codon in *URA3* more frequently and grow on uracil-lacking media. DNA is extracted from colonies grown on uracil-lacking media, and used in subsequent round of TadA-TadA* library generation and enrichment. Iterating the process yields increased numbers of colonies on uracil-lacking plates (rounds 1–5) as TadA-TadA* variants with enhanced A/T mutagenesis enrich. **b** A/T targeted mutation rate in strain CSY1256 measured by reversion of a TAA stop codon in *URA3* and quantified using the Falcor algorithm. The on/off-target mutation rates are compared for parent enzyme BE7.10 (gray, pCS4332) and newly engineered variant yeTadA1.0 (blue, pCS4333). Consensus mutations resulting from 5 rounds of evolution (in blue) with pre-existing TadA* mutations (gray). Error bars represent 95% confidence intervals measured from eight independent cultures induced with 0.2% galactose. **c** Mutation frequencies for different substitution types (left corner of each plot), plotted by position along *URA3* for strains with the engineered yeTad1.0 incorporated (YHM5, blue lines; YHM6, black lines) and control lacking yeTadA1.0 (YHM2, yellow lines) and control lacking T7 targeting (YHM2.P$_{T7}$KO, gray lines) when induced with 100 nM β-estradiol for 16 h. In order to match YHM2 and YHM2.P$_{T7}$KO, data from YHM2.P$_{T7}$KO is off-set by 23 bp, accounting for the T7 promoter. Mutation data generated by next-generation sequencing (NGS) at *URA3* as described in "Methods". **d** Fraction of base substitutions occurring at each base in the mutator strain YHM6 as measured by NGS; substitutions occur across all four nucleotides. Mutation frequencies were calculated in duplicate from NGS data across bases 1000–1200 at *URA3* in YHM6 and for each substitution type the background sequencing error frequencies (calculated from YHM2.P$_{T7}$KO) were subtracted to account for the contribution of sequencing error. Source data are available as a Source data file.

determine specific GOF mutations. Of 16 sequenced mCherry-positive colonies, all contained a N13Q mutation resulting from T > C substitution of the TAA stop codon. Our results demonstrate that TRIDENT can introduce mutations far from the T7 initiation site at distances exceeding the range of other recent in vivo mutagenesis systems utilizing Cas9 for targeting[1].

We applied the TRIDENT platform to evolve more complex protein phenotypes without prior knowledge of required mutations, by engineering a red-shifted mCherry variant. A classic modification to spectral properties of fluorescent proteins is shifting the emission spectrum, which can allow for new applications such as in situ imaging[22]. A low-copy plasmid encoding an mCherry expression cassette with upstream P$_{T7}$ for targeting (pCS4335) was introduced into strain YHM6. Mutagenesis was induced with 10 nM β-estradiol for 16 h. We assessed the diversity of the library prior to sorting by sequencing mCherry from individual colonies and found that mutations were spread throughout the gene with an average mutation rate of $1.0 \times 10^{-3}$

s.p.b. across all four bases in 98 sequenced colonies (Fig. 4d). From this initial library, we enriched for red-shifted variants by sorting the 1% of cells having the highest ratio of 615:660 nm emission and outgrowing for 24 h without additional mutagenesis before sorting again. After four rounds of enrichment, we observed a shift in the population of cells off of the diagonal and plated these cells to isolate single red-shifted variants. Of 12 sequenced colonies, all non-parent variants contained a single F23S mutation resulting from a T > C substitution of TTC to TCC. To assess the degree of red-shift, wild-type mCherry (pCS4337) and mCherry-F23S (pCS4338) were expressed in *E. coli* and the emission spectra analyzed using a plate reader. The data indicate that mCherry-F23S is 5 nm red-shifted compared to wild-type mCherry[23] with an emission max of 615 nm compared to 610 nm (Fig. 4c).

Finally, we demonstrated that the TRIDENT platform can be utilized in a semi-continuous evolution format by evolving resistance to pyrimethamine. The dihydrofolate reductase

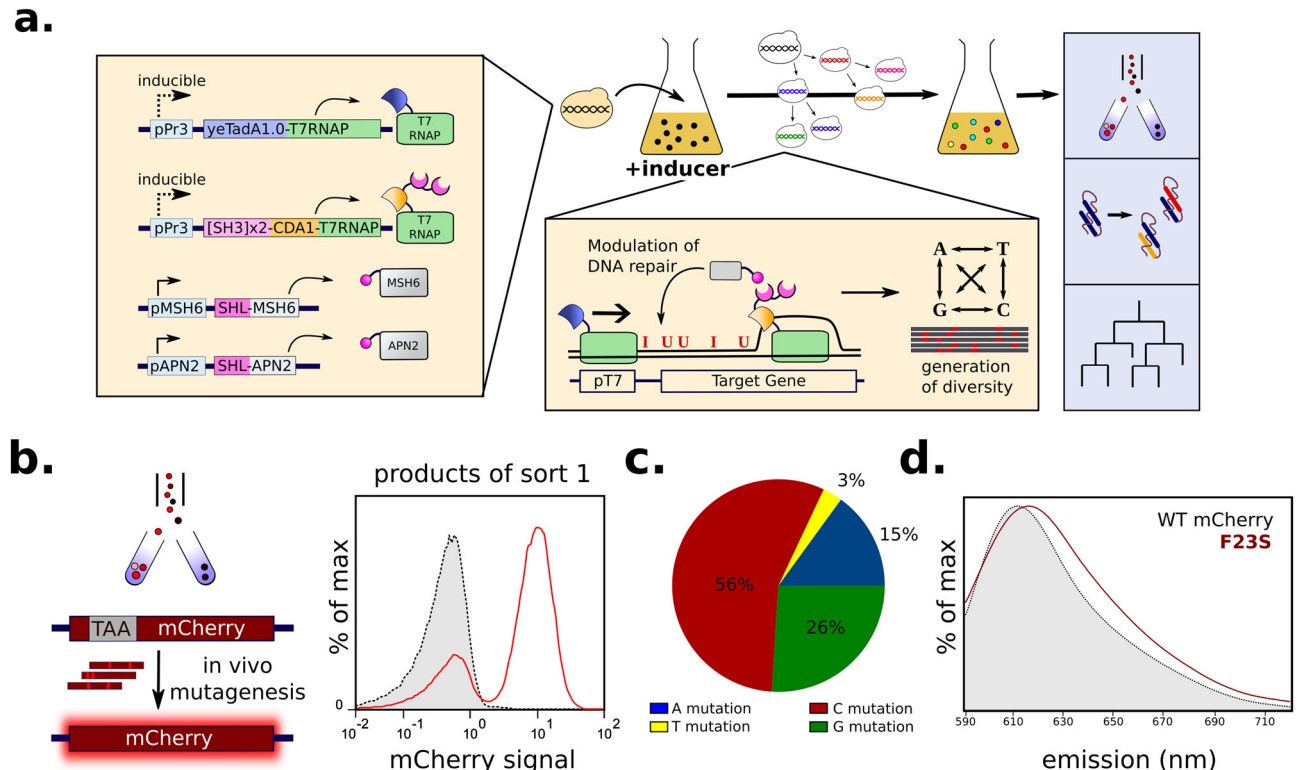

**Fig. 4 Overview and application of TRIDENT to protein evolution challenges. a** Schematic of primary modifications made in the most advanced TRIDENT strain platform, YHM6 (left). When YHM6 is grown with mutagenesis inducer, in vivo mutagenesis creates a library of variants for applications including cell sorting, protein engineering, or lineage tracing (right). **b** TRIDENT applied to a cell sorting application in which a TAA stop codon in mCherry is corrected with in vivo mutagenesis and mCherry-positive cells are collected by FACS (left). Strain YHM6.mCherry.TAA was grown with 10 nM β-estradiol for 16 h and mCherry-positive cells were sorted once yielding a 0.14% mCherry-positive population. The mCherry-positive post-sort population (red) was reanalyzed demonstrating correction of the stop codon. The non-fluorescent presort population of YHM6.mCherry.TAA cells is shown for comparison (gray). **c** YHM6 generates mutations at all four nucleotides on a plasmid-based copy of mCherry (pCS4335) with $P_{T7}$ upstream for mutational targeting. YHM6 + pCS4335 was grown with 10 nM β-estradiol for 16 h and plated. Ninety-eight presort single colonies were picked and mCherry sequenced off the plasmid in order to calculate the fraction of base substitutions occurring at each base of mCherry. **d** TRIDENT can be applied to evolve more complex protein properties such as spectral properties of a fluorescent protein. TRIDENT-evolved mCherry variant, mCherry-F23S, is red-shifted by 5 nm. The emission spectra of evolved mCherry red-shift mutant (pCS4338) compared to wild-type mCherry (pCS4337) expressed from *E. coli*, was measured at excitation of 561 nm. mCherry had max emission of 610 nm and mCherry-F23S had max emission of 615 nm. Source data are available as a Source data file.

(DHFR) of *Plasmodium fulciparum* is competitively inhibited by pyrimethamine[24]. We designed a yeast strain, YHM7, that cannot grow in high concentrations of pyrimethamine by knocking out the essential, endogenous DHFR gene and replacing it with a *P. falciparum* DHFR gene expression cassette flanked by an upstream T7 promoter and downstream T7 terminator. To demonstrate the robustness of the platform, mutagenesis was carried out in 90 experimental replicates at induction concentrations of 2 nM β-estradiol and 100 nM β-estradiol. The concentration of pyrimethamine was increased stepwise in five total passages from a starting concentration of 100 μM to a final concentration of 3 mM, the solubility limit of pyrimethamine. Passages required 1–3 days to reach full saturation for a majority of experimental replicates. After five passages totaling 11 days of mutagenesis, 177 of 180 experimental replicates grew to saturation, demonstrating full resistance to pyrimethamine.

We showed that TRIDENT generated mutations of all four possible nucleotides during the semi-continuous evolution experiment. Each of the fully grown experimental replicates was uniquely barcoded and sequenced. Mutations were considered significant and further investigated if they were present in at least 10% of reads or if they were present in at least four experimental replicates. Four significant coding mutations were found: D54N, Y57H, Q237*, and N277D (Fig. 5a). Two of these mutations,

D54N and Y57H, have been identified in previous studies as impacting the interaction between pyrimethamine and PfDHFR[24,25], and the most common mutation, D54N, was present in over 85% of individual sequences. Analysis of the crystal structure[26] of the enzyme showed that the D54 residue is within 3 Å of the pocket where pyrimethamine binds (Fig. 5b), suggesting that this mutation enables pyrimethamine resistance by directly modulating the binding pocket.

All four significant coding mutations were tested to determine their individual and combined impacts on the growth rate of the yeast in media containing 3 mM pyrimethamine. Each coding mutation was tested individually and in combination with the D54N mutation (Supplementary Figs. 17 and 18). Wild-type PfDHFR was used as a control for growth rate comparison. All yeast with PfDHFR variants containing the D54N mutation grew significantly faster than yeast with wild-type PfDHFR. Two other mutations (Y57H and Q237*) appeared to have a positive impact on growth rate, but the improvement compared to wild-type was not statistically significant in the context of this experiment. The significant enrichment of the D54N mutation and the validation that D54N significantly improves the growth rate of yeast in the presence of pyrimethamine show that TRIDENT successfully enriched for mutations in the target gene that improve yeast growth rate.

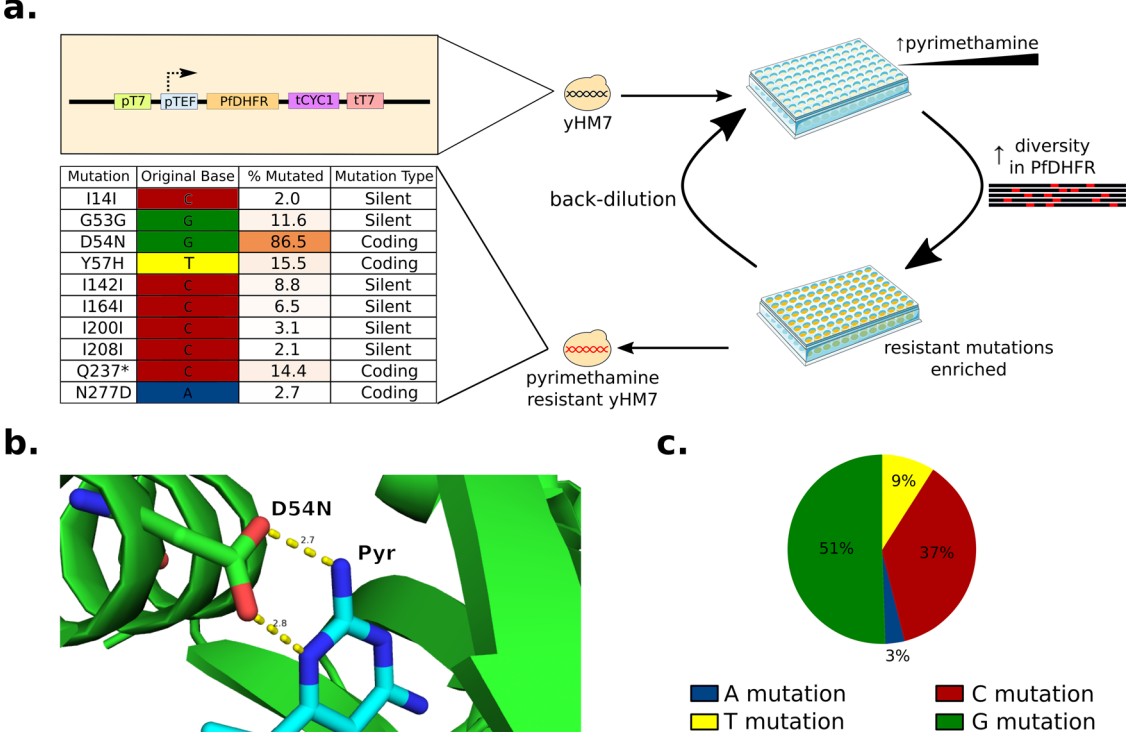

**Fig. 5 Overview of workflow and results of engineering PfDHFR for pyrimethamine resistance. a** Schematic of overall strategy for engineering pyrimethamine resistance in yeast. Modifications to YHM7 (top left) target TRIDENT mutations to the PfDHFR gene expression cassette. Replicates of YHM7 were grown in a 96-well plate containing pyrimethamine and inducer. Diversity generation and enrichment of beneficial mutations occur simultaneously as the cells grow to saturation. After reaching saturation, replicates are back-diluted into an increased amount of pyrimethamine until full resistance to the drug has been achieved. Pyrimethamine resistant samples were sequenced and results are shown in the table (bottom left). This schematic was created using Servier Medical Art images, which are licensed under a Creative Commons Attribution 3.0 Unported License; https://smart. servier.com. **b** PfDHFR (green) binding pocket crystal structure with bound pyrimethamine (light blue). The D54 residue on PfDHFR is shown in detail. The potential interactions between D54 and pyrimethamine are highlighted by distance measurements given in angstroms. **c** Distribution of all mutations found in PfDHFR gene when 177 experimental replicates were sequenced after five passages in pyrimethamine. Mutations of all types possible in the TRIDENT system were found. The distribution of base mutation types is different from other experiments using TRIDENT on other genes with different selection pressures (Figs. 3d and 4c) and reflects the dominance of the D54N mutation.

## Discussion

Development of robust systems for in vivo random mutagenesis would enable applications as diverse as lineage tracing, continuous evolution, and rapid protein engineering. In the past decade a handful of systems have been developed for in vivo mutagenesis[1–8] each with constraints around specificity, mutation rates, and control of mutational diversity. In this work, we develop TRIDENT, an engineered targeted in vivo mutagenesis platform capable of achieving mutation rates and base diversity sufficient for directed evolution of protein function.

We built TRIDENT to mimic the polymerase-based targeting of B-cell SHM by utilizing T7 RNAP as a means of targeting diverse DNA modifying enzymes to a target gene. To develop TRIDENT, we first engineered targeting of a cytosine deaminase by fusion of PmCDA1 to T7 RNAP and demonstrated controlled induction of targeted mutation rates over a 100-fold range (strain YHM1) using a synthetic inducible promoter combined with disruption of Uracil DNA Glycosylase (UNG). Characterization of the mutation spectra occurring in YHM1 showed that over 99% of induced mutations are C > T transitions, consistent with past work in which PmCDA1 was over-expressed in UNG-deficient yeast[27].

In order to enhance the rates of A/T/G mutagenesis we expanded the TRIDENT platform to target DNA-repair factors implicated in B-cell SHM. We co-localized yeast homologs of 12 DNA-repair factors involved in SHM to T7 RNAP using modular SHL/SH3 domains and measured resulting changes in A/T mutation rate. Simultaneous targeting of a mismatch and base

excision repair protein (Apn2p and Msh6p) increased rates of A/T mutation by 5-fold and increased rates of G mutagenesis by at least 5-fold. Recent work has shown that targeting UNG using an UNG-Cas9 fusion can increase base diversity[28]. However, in our system overexpression or targeting of UNG served only to reduce the overall on and off-target mutation rates. We suggest differences in host (mammalian vs. yeast) repair mechanisms and localization strategies (Cas9 vs. T7 RNAP) is responsible for our differing observations.

Finally, in order to balance the rates of A/T and C/G mutagenesis, we expanded the TRIDENT platform to target an engineered adenosine deaminase, yeTadA1.0, which deaminates adenosine to inosine resulting in A-to-I-to-G transitions. We evolved a previously engineered TadA variant[21] for high activity in our system increasing the targeted A/T mutation rate by 270-fold. Expansion of the TRIDENT platform by incorporation of the engineered yeTadA1.0-T7 RNAP fusion enables precise, targeted mutagenesis at A/T and functions alongside C/G mutagenesis generated with PmCDA1 and SHL targeting of repair factors. As a reference for application of TRIDENT, our data suggest that the platform generates 1 mutation per kilobase of sequence per sixteen hours of growth. Our experimental design for TRIDENT-directed evolution experiments used a single T7 terminator; however, Moore et al. showed that four or more T7 terminators may be needed to efficiently terminate T7 RNAP transcription and prevent downstream mutagenesis for experiments performed in highly sensitive chromosomal regions[8].

We showed how TRIDENT can be applied to efficiently evolve protein properties which require screening or selection. In developing TRIDENT, we routinely demonstrated the ability of TRIDENT to function in simple selections by correcting or producing point mutations in the URA3 which allowed cells to survive in culture media lacking uracil. The complete TRIDENT system was subsequently demonstrated to be compatible with FACS screening of the fluorescent protein mCherry in which we both reverted a stop codon in mCherry and evolved an mCherry variant displaying a red-shifted emission spectrum. Last, multiple rounds of selection with increasing selection pressure were performed to evolve variants of PfDHFR resistant to pyrimethamine. After five rounds of selection, 177 of 180 experiments were able to grow to saturation at the highest soluble concentration of pyrimethamine. Sequencing of the experimental replicates revealed that the D54N mutation was heavily enriched. Validation experiments confirmed that the D54N mutation resulted in a faster growth rate.

Together, these experiments demonstrate the flexible, rapid, and robust nature of the TRIDENT system. TRIDENT is biased toward generation of C > T mutations, but in practice, we still identified significant T > C mutations (as in the F23S mutation causing the red-shift in mCherry) and G > A mutations (as in the D54N mutation found in pyrimethamine resistant variants of PfDHFR). These results indicate that the mutation rate of each part of the TRIDENT system is high enough to be able to screen or select for beneficial mutations of these types. The high mutation rate also results in very rapid evolution of traits. The red-shifted mCherry variants were found after a single day of induced mutagenesis. The five rounds of selection yielding pyrimethamine resistant yeast in the PfDHFR experiment required 11 days of induced mutagenesis which is more than twice as fast as other in vivo evolution experiments using the same PfDHFR framework[4,29]. In addition, pyrimethamine resistance was evolved at two different inducer concentrations (2 and 100 nM β-estradiol) with 90 replicates each. In total, 177 of 180 experimental replicates grew to saturation in 3 mM pyrimethamine within 3 days of growth, showing the robustness of the TRIDENT system.

The TRIDENT system is uniquely effective, easy to use, and comprehensive. TRIDENT's mutation rate is both high enough to produce quick mutagenesis results and also tunable for use with more sensitive proteins. The system requires minimal cloning and genomic modification, as the only requirement to target mutagenesis is the insertion of a T7 promoter. In addition, TRIDENT is the only system able to make both A/T mutations and C/G mutations at similar rates across multiple-kb of DNA, though the system shows bias towards transition mutations that limitation can be addressed through further tuning of the DNA-repair modifying pathways described in this work. The ability to make mutations from all four base pairs increases the evolutionary space that can be explored. TRIDENT generates rapid and meaningful diversity simply by the addition of a mutagenesis inducer and subsequent cell growth, providing a platform for accelerating protein engineering and experimental evolution.

## Methods

### Chemicals, media, and strain cultivation
Difco yeast nitrogen base without amino acids and ammonium sulfate (YNB), Bacto peptone, Bacto yeast extract, Luria Broth (LB), LB agar, dextrose, and galactose were obtained from BD (Becton, Dickinson and Company). PBS tablets were obtained from VWR. BSA was obtained from Cell Signaling Technologies. Kanamycin monosulfate and ampicillin were obtained from Sigma Chemicals. G418 (sc-29065A) used for yeast integration selections was obtained from Santa Cruz Biotech. Amino acid dropout media was obtained from Takara Bio (product #630400–#630431). 5-FOA powder and Frozen-EZ transformation kit for yeast were obtained from Zymo Research.

*E. coli* strains were selected on LB agar plates with 50 mg/L kanamycin, 50 mg/L ampicillin, and grown in LB liquid media with the appropriate antibiotic. Yeast 10x drop out (DO) supplement was prepared from Takara as a synthetic complete supplement with desired dropout component omitted. *S. cerevisiae* strains were selected on YNB-DO (0.17% yeast nitrogen base, 0.5% ammonium sulfate, 2% dextrose, and 1x DO) agar or on YPAD (1% yeast extract, 2% peptone, 80 mg/L adenine hemisulfate, and 2% dextrose) agar. For selections, plates were supplemented with 200 mg/L G418 or 400 mg/L 5-FOA. Yeast were grown in selective YNB-DO media or in YPAD media.

### Plasmid construction
Strains and plasmids used and constructed in this work are described in Supplementary Tables 3 and 4. *E. coli* strain TOP10 (Life Technologies) was used for all cloning and amplification of plasmids. Plasmids were recovered using Econospin columns (Epoch Life Sciences) according to manufacturer's instructions. All oligonucleotides were synthesized by the Stanford Protein and Nucleic Acid Facility. All PCR was carried out using Q5 DNA Polymerase (New England Biolabs) and all restriction enzymes, T4 DNA ligase, and deoxynucleotides were purchased from New England Biolabs. Heterologous gene sequences were cloned from previously published plasmids, obtained from Addgene, or synthesized by IDT. Cloning was exclusively performed using Gibson assembly[1] followed by transformation into *E. coli* or by gap repair directly into yeast, or direct genomic integration. gRNA integration plasmids were constructed from addgene plasmid pCAS, which was a gift from Jamie Cate (Addgene plasmid # 60847)[2,3]. pCAS was modified by Gibson assembly to create SpCas9 expression vector pCS3410, which was digested with PacI followed by Gibson assembly of each gRNA fragment. All gRNA primer sequences used in this work are listed in Supplementary Table 4. All full-length protein sequences are available in Supplementary Table 5.

DNA purification columns, Zymoclean gel DNA recovery kit, and yeast genomic DNA prep kit (D2002) were all obtained from Zymo Research. Next-gen sequencing DNA prep was carried out using the Nextera XT DNA Library Preparation Kit (24 sample kit) and corresponding adapter primers were obtained from Illumina. All routine sequencing was performed using Elim Biopharmaceuticals.

### Yeast plasmid transformation and integrated strain construction
*S. cerevisiae* strain CEN.PK2-1Dα was used as the base strain for in this work with the exception of the evolution of yeTadA1.0 adenosine deaminase, which was performed in W303α as noted in Supplementary Table 1. Characterization of TadA and engineered yeTadA1.0 was performed in a CEN.PK2α as noted in the figures and Supplementary Table 1. All yeast plasmid transformations were performed using EZ transformation kit from Zymo Research according to manufacturer's instructions, unless otherwise specified. Cloning was performed using a combination of Gibson assembly and DNA gap repair in the yeast strains specified in Supplementary Table 1. Scar-free yeast integrations were carried out as previously described[2], by co-transformation of gRNA/Cas9 expression plasmid targeting the edit site, along with repair DNA to be integrated genomically. Briefly, expression cassettes or gene modifications were amplified using primers with 15–40 bp overhangs for gap repair in yeast. Each assembly was flanked by an integration homology region of 30–50 base pairs. gRNAs were generated at a target region such that integration of the desired modification would result in disruption of the gRNA binding site. Typically, 100 ng of purified PCR product was introduced with 300 ng of gRNA plasmid using the Zymo EZ Kit for transformation.

### Fluctuation analysis
Fluctuation analysis for determination of URA3 loss-of-function (LOF) mutation frequency was carried out at a genomic copy of URA3 (with and without $P_{T7}$ targeting sequence) integrated at *Lyp1*. Cells were grown for 16 h in SC + 2% raffinose media either lacking uracil, uracil and tryptophan, or uracil and leucine depending on the autotrophic markers required to maintain plasmids. Cultures were then diluted 1:10,000 into inducing media and split into 50 μL volumes in 96-well plates. Inducing media consisted of 2% raffinose with 0.2% galactose for galactose-based induction or 2% dextrose with 100 nM β-estradiol unless otherwise specified. Eight independent wells were used per strain unless otherwise specified. Plates were sealed and incubated at 30 °C unless otherwise specified for 14–40 h depending on the mutation rate being measured. Each well was resuspended by pipetting and 30 μL plated onto SC plates containing 5-FOA at 500 mg/L. The remaining volumes were pooled between replicate wells and plated onto YPD at different dilutions for titering. Colonies on 5-FOA and titer plates were counted after 2 and 3 days, respectively.

Fluctuation analysis for determination of URA3 gain-of-function (LOF) mutation frequency at TAA stop codon was carried out at a genomic copy of URA3 (with and without $P_{T7}$ targeting sequence) integrated a *Lyp1*. *Lyp1*::URA3 was edited with gRNA g834 to yield a TAA at E8 in URA3 (Supplementary Table 4). Cells were grown for 16 h in SC + 2% dextrose media and cultures were then diluted 1:10,000 into inducing media and split into 150 μL volumes in 96-well plates. 100 nM β-estradiol was used unless otherwise specified and eight independent wells were used per strain unless otherwise specified. Plates were sealed and incubated at 30 °C unless otherwise specified for 20–48 h depending on the mutation rate being measured. Plates were spun down, pellets washed once with sterile water, and each well was resuspended in 60 μL with 50 μL plated onto uracil-lacking plates. The remaining volumes were pooled between replicate wells

and plated onto YPD at different dilutions for titering. Colonies on YPD and uracil-lacking plates were counted after 2 and 3 days, respectively.

*URA3* loss and gain-of-function rates were calculated using the Falcor algorithm Ma-Sandri-Sarkar Maximum Likelihood Estimator (MSS-MLE) method[4] written in a custom Python script available on GitHub (see the sections "Methods" and "Code availability"). MMS-MLE also provided for calculation of 95% confidence intervals. The partial plating correction was not applied, as the calculated loss and gain- of-function mutation rates are proxy estimates for genomic per-base mutation rates, and only used comparatively in our analysis.

**Next-gen sequencing and analysis for characterization of TRIDENT system.** Cells were grown for 16 h in SC + 2% dextrose media with either synthetic complete amino acids lacking uracil or uracil and tryptophan for integrated mutator constructs and plasmid-based constructs, respectively. Cultures were back-diluted 1:100 into 3 mL of SC + 2% dextrose media containing β-estradiol inducer at 500 nM or otherwise indicated concentrations, and grown for 20 h. Cells were harvested, genomic DNA extracted using the Zymo genomic kit, and the target region was PCRed out using the primers GAGAGCAATCATCCCTATATGCC and GTGTAGCGGTGTACCAAAAGTT. PCR products were processed using the Nextera kit and samples were sequenced on HiSeq and MiSeq machines at 2 × 150.

NGS data were aligned to reference sequences using Bowtie with the following parameters: *bowtie $filename -5 4 -3 4 -n 3 -e 450 -l 16*. Bowtie output was parsed using custom Python code available on GitHub (see the sections "Methods" and "Code availability"). Briefly, base calls having Quality Scores <35 were removed from the dataset and the remaining calls were counted at each position to determine occurrence frequency of A, T, C, and G at each site. The frequency of each base at a given position was calculated as a ratio of observed occurrences of each individual base divided by total occurrences of A, T, C or G at that position, as shown below. $N_A$ is the number of counts of base $A$ at position $i$ in the DNA sequence.

$$\text{Frequency}_{A \text{ mutation}, i} = \text{Number}_{A, i} / \left[ N_{A, i} + N_{T, i} + N_{C, i} + N_{G, i} \right]$$

Trend line curves as in Fig. 1d were calculated as the unweighted moving average of the frequencies of the adjacent 10 base positions on either side. Individual base frequencies (e.g., frequency$_{A \text{ mutation}, j}$) greater than three standard deviations from the mean frequency at position $i$ were removed from the moving average to reduce the influence of extreme outliers occurring from N-mer sequences (3-sigma rule for outlier detection)[5].

**Base editor evolution.** In order to evolve TadA-TadA*-T7 RNAP as a mutator in yeast, libraries of TadA-TadA* were created and screened using a growth-based assay to isolate variants with improved mutagenic activity. This was done by using a *URA3* variant with a premature stop codon (TAA) that could be corrected through action of A:T → G:C editing. Yeast containing TadA variants were grown on uracil-lacking media, and only those cells with gain-of-function mutations at TAA contained a functional copy of *URA3* to complement growth. Given a background mutation rate of $10^{-8}$ for TAA reversion in wild-type yeast, an enrichment of ~1000-fold was expected in each round of evolution and selection.

**Base editor evolution: strain construction.** Yeast strain CSY1256 was used with the following modifications: (1) a *URA3* gene containing a premature TAA stop codon and a T7 promoter upstream of *URA3* was integrated at *Lyp1* using gRNA g101, (2) the native ura3-1 allele was deleted using gRNA g834 and insertion of *HIS3*, and (3) the native *Mag1* gene was knocked out through insertion of a premature ADH1 terminator using gRNA g791 (Supplementary Table 4).

**Base editor evolution: library construction.** DNA encoding BE7.10 deaminase domain (TadA-TadA*) was synthesized as an IDT gene block codon optimized for yeast expression. TadA-TadA* was gap repaired into plasmid pCS4314 digested with PstI and BamHI, and the subsequent plasmid was termed pCS4332. An error-prone library of the TadA domains was made using pCS4332 as template, and mutations were introduced using Taq polymerase and TriLink Biotechnologies mutagenic dNTPs (Item #2748 and #2746) according to the manufacturer's instructions. Three separate PCR reactions were performed at 20 cycles with varying concentrations of equimolar nucleotide analogs according to manufacturer instructions and products from each mutagenic PCR were amplified further using Taq polymerase. Amplified DNA was column-purified. The PCR primers used were CTGGGGTAATTAATCAGCGAAGCG and TAGCTATGTTAATCGTGT TAACCTGATTGTCATCGTCTTTGTAATCAAATATCATGATC, and provided >50 bp of sequence overlap with the subsequently digested vector. Cut vector was prepared from pCS4332 via digestion with PstI and BamHI overnight to ensure complete digestion and then column-purified. Library transformation into CSY1256 was carried out as previously described[6] using the mutant PCR products and digested vector. Library size for each round was estimated as between $10^6$ and $10^7$ transformants by dilution plating.

**Base editor evolution: library screening.** Transformed CSY1256 was outgrown for 24 h in 100 mL of YNB dextrose media lacking tryptophan to an $OD_{600}$ of 10 to obtain $10^3$ to $10^4$ copies of each clone. Half of the cells were pelleted and

resuspended in 150 mL YNB lacking tryptophan but with 2% added galactose to induce expression of the TadA-TadA*-T7 RNAP library. Cells were grown in inducing media for 16 h, pelleted, and resuspended in 150 mL YNB dextrose lacking tryptophan and uracil to amplify any cells which gained *URA3* function from TAA-reversion mutations. After another 16 h outgrowth, 1 mL of culture was plated onto -UT plates, which were grown for 2 days at 30 °C. Several colonies were picked for analysis of TAA reversion and TadA-TadA* mutations and the remaining colonies were scraped from the plate and plasmid was extracted using the Zymo yeast plasmid extraction kit according to manufacturer's instructions. Extracted plasmid DNA was used as template in the next round of mutagenic PCR serving to add additional mutations and enrich for any improved TadA-TadA* variants.

**Growth rate measurement for assessing cellular fitness.** Yeast strains were inoculated in triplicates into a 400 μl culture of synthetic complete media in 96-well plates and grown overnight to saturation prior to 100-fold back dilution into a 400 μl culture of synthetic complete media, supplemented with 0, 5, or 100 nM β-estradiol. The optical density at 600 nm (OD600) of the culture was then measured at 4, 8, 12, 16, 20, and 24 h of growth using Tecan Safire2 plate reader (10 s orbital shaking prior to measurement, final absorbance collected from an average of 15 laser readings). For each time point, 100 μl of culture was sampled and transferred back to the culture plate for further growth after OD600 measurement. The mean and standard deviation of OD600 was then calculated from the triplicates of each strain at each time point. For calculating growth rates from OD measurement, the following equation was used:

$$\text{Growth rate} = \ln\left(OD600_{(t=16 \text{ h})}/OD600_{(t=4 \text{ h})/12}\right)$$

**mCherry library screening.** mCherry libraries were constructed as follows: cells were grown for 16 h and then diluted 1:100 into 20 mL and induced with 10 nM β-estradiol for 16 h at 25 °C. Cells were pelleted and resuspended in ice-cold PBS with 1% BSA prior to sorting. Cell sorting/flow cytometry analysis for this project was done on instruments in the Stanford Shared FACS Facility. For stop codon reversion experiments with YHM6.mcherry.TAA, $10^7$ cells were sorted with a BD Aria II Cell Sorter using a 561-nm laser and a 616/23 nm emission filter. For red-shift experiments with YHM6 and pCS4335, $10^7$ cells were sorted each round using a 561-nm laser and the following emission filters: 616/23 nm, 660/20, and 785/62. We sorted ~1% of cells each round farthest off the diagonal of FACS plots of 616/ 23 × 660/20 and 616/23 × 785/62.

**mCherry protein expression and characterization.** Red-shifted and wild-type mCherry variants were PCRed amplified from yeast vectors (pCS4335, pCS4336) and Gibson assembled into pET28 vector (pCS3306) which was digested with XbaI and XhoI, making protein expression vectors pCS4338 and pCS4337. Protein expression was carried out in BL21 cells for 24 h at 25 °C with 1 μM IPTG. Cells were harvested and 10 mg of cell pellets then resuspended into TE buffer in a black 96-well plate. Fluorescent protein emission profile was characterized using a Molecular Devices Spectral Plate Reader with the following settings: excitation: 561 nm, step: 1 nm.

**Pyrimethamine resistance evolution, sequencing, and validation in PfDHFR.** The PfDHFR amino acid sequence used in this study was chosen based on the availability of a crystal structure (IJ3J)[8]. The gene was then codon optimized for expression in Saccharomyces cerevisiae using the JCAT codon optimization and synthesized using Twist Bioscience. The synthesized gene was amplified by PCR and Gibson assembled into an expression vector including a T7 terminator after the yeast terminator (PCS 4601). This gene expression cassette was then incorporated into the genome of the YHM6 strain at the URA site (which already contains a T7 promoter). Evolutionary passages were grown in 400 μL of synthetic complete in 96-well plates at induction concentrations of 2 and 100 nM β-estradiol. Pyrimethamine concentrations increased with each passage as follows: 100 μM > 500 μM > 1 mM > 2 mM > 3 mM. Synthetic complete was supplemented with 1–3% DMSO to solu-bilize higher concentrations of pyrimethamine. The pH of the media was adjusted to 4.5 to ensure pyrimethamine solubility of the 3 mM pyrimethamine. Plates were back-diluted at a 1:100 ratio when >90% of wells were overgrown.

The portion of the genome containing the PfDHFR gene was amplified by colony PCR and a second PCR reaction was performed to attach barcodes corresponding to the well from which the amplicon originated (Supplementary Fig. 18). Samples were then sent to the Chan Zuckerberg Biohub and sequenced using a Pacbio Sequel II system. Sequencing data was initially uploaded to the Galaxy web platform and the public server at usegalaxy.org was used for the initial processing of reads[10]. A total of 2.3 million raw reads were obtained and filtering for quality, length, and front and back barcode fidelity resulted in roughly 486,000 usable reads. The sequences from each well were then aligned using MAFFT flavor FFT-NS-1[11]. A custom python script was used to eliminate blanks from the aligned sequences and quantify the distribution of bases at each position in the PfDHFR gene. This provided an overall view of the mutation frequencies at each position for each of the 177 experimental replicates that grew to saturation in 3 mM pyrimethamine. Mutations were considered significant and analyzed if they appeared in at a global frequency of at

least 10% or if they were present in at least 4 of the 177 replicates (Supplementary Fig. 19). Four significant coding mutations were identified. To validate their impact on growth, PfDHFR variants were expressed via pCS4601 in strain CSY1321 (Supplementary Table 1). Each coding mutation was tested individually and the lower frequency mutations were tested in combination with D54N. Variants were grown in 2 mL synthetic complete without tryptophan and with 3 mM pyrimethamine. Each variant was grown in triplicate and $OD_{600}$ measurements were recorded at 4-h intervals for 36 h (Supplementary Figs. 16 and 17). Statistical significance of differences in growth rate was determined by comparing the optical density of different variants at each timepoint using the *t*-test.

**Statistics and reproducibility.** No prior estimation of requisite sample size was performed. All presented data represent measurements from sample sizes of at least three biological replicates, where biological replicates (as defined in "replication" below) represent independently grown microbial cultures. Since (i) engineering was performed at the cellular level, (ii) mutation rate is a bulk measure of cellular population, and (iii) each assayed microbial culture represents a large population of individual cells, three biological replicates were sufficient for reliable measurement of changes in mutation rate at the population level. All measurements of mutation rate, growth rate were repeated on independent biological samples (i.e., independent microbial cultures or strains). All attempts at replication of results presented in this study were successful. All replicates performed in this study were biological replicates, rather than technical replicates, and represent independent data points. For example, replicate microbial cultures were grown in separate containers and assayed independently from one another. Fluctuation analysis results, FACS, NGS, and viability growth rate, experiments were replicated at least three times.

For mutation rate measurements using fluctuation analysis, single data points are not shown, as the method used to calculate mutation rate uses $P_0$ statistics and thus plotting individual data points would not be appropriate. There was high variability in the measurements of TAA mutation rates with SHL-fusion strains, as shown in Fig. 2c, and so for this experiment $N$ was increased to $N = 24$, the largest number allowing the experiment to still be carried out in 96-well plates. The yeast continuous evolution experiments were carried out in a highly parallel format over the course of weeks and growth results from individual wells replicated by cloning of individual sequences.

**Reporting summary.** Further information on research design is available in the Nature Research Reporting Summary linked to this article.

## Data availability
Data supporting the findings of this work are available within the paper and its Supplementary Information files. Raw NGS data is publicly available through NCBI under accession number PRJNA701053. Protein and primer sequences are listed in supplementary materials. All yeast strains and plasmids described in this work are available upon request. The engineered yeTadA1.0 enzyme has been deposited and is available through Addgene, in anticipation of specific requests for this material (Addgene #137735, yeTadA1.0-100AA linker-T7 RNAP fusion protein). Source data are provided with this paper.

## Code availability
Code used to analyze NGS data, generate sequencing plots, and execute the MLE method for fluctuation analysis is available on GitHub at https://GitHub.com/aaroncp1an0/TRIDENT.

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

## Acknowledgements
We thank Dr. Benjamin Kotopka for valuable feedback in the preparation of the manuscript. This work was supported by the National Institutes of Health (grant to C.D.S., AT007886), CZ Biohub (investigator grant to C.D.S.), and Agilent Foundation (fellowship to A.C). We thank the Genome Sequencing Service Center by Stanford Center for Genomics and Personalized Medicine Sequencing Center, supported by the grant award NIH S10OD020141.

## Author contributions
A.C., C.D.S., O.K.J., and J.T.S. conceived of the project, analyzed the results, and wrote the manuscript. A.C. and O.K.J. designed and performed all experiments and analyzed the data. O.K.J. engineered yeTadA1.0 and carried out PfDHFR evolution. D.K. constructed select strains, performed mCherry evolution, and growth curves.

## Competing interests
The authors declare no competing interests.

**Additional information**

