## [Peer Review File · Nature Communications]

Reviewers' Comments:

Reviewer #1:

Remarks to the Author:

In this manuscript, Cravens A and Jamil O, et al. used a T7 RNA polymerase (T7 RNAP)-based approach to induce targeted DNA mutagenesis in yeast. They demonstrated the capacity of the TRIDENT system by correcting or producing point mutations in the URA3 gene, evolving the mCherry gene and the PfdHFR gene. Although similar approaches have been recently reported, I appreciate the authors' efforts to diversify the mutational spectrum of this system through co-localizing yeast DNA repair factors involved in SHM to T7 RNAP using modular SHL/SH3 domains and evolving a yeast TadA enzyme for A/T and C/G mutagenesis.

My main concern is about the off-target effects of TRIDENT. In the current study, the off-target mutation rate was measured via the fluctuation assay in a strain lacking the T7 promoter upstream of the URA3 gene. Have the authors looked at the mutation rates of other genomic regions other than URA3? On a related note, although it may be less of an issue for the purpose of directed evolution, have the authors examined if expressions of the TRIDENT system, especially in the presence of multiple engineered enzymes negatively impact the health status of the yeast cells?

Second, I think for the purposes of continuous directed evolution, it would be very important to show multi-step fixation of mutations, and/or longer timescales of directed evolution of proteins with TRACE.

Some minor issues:

1. Page 3, line 78-81, although it is true that the TRACE system has a notable bias to C/G transition mutations, the authors' claim that it lacks tunability of overall mutation rate may not be accurate. It has been shown that the overall mutation rate can be tuned by using cytidine deaminase with different enzyme activity as well as using T7 RNA polymerase variants with different DNA processivity and elongation rate.
2. The authors should also acknowledge the work by Moore CL et al (J. Am. Chem. Soc. 2018, 140, 37, 11560–11564) and Álvarez B et al (bioRxiv, doi: <https://doi.org/10.1101/850974>) in which a similar T7 RNA polymerase-based design was used for in vivo mutagenesis.
3. Looking at the distribution of base substitution rates at the URA3 locus after the T7 promoter (e.g., Fig1D, Fig2B and Fig3C), there seems to be a change in base substitution rate as a function of the distance to the T7 promoter for certain types of substitution. Is it true? Have the authors quantify this?
4. How long is the PfdHFR gene? I'd like to see some of the representative alignments of the Pacbio data on the evolved PfdHFR gene sequence. In particular, I'm interested to see if the T7 terminator at the end of the PfdHFR gene was able to stop the editing events since Moore CL et al (J. Am. Chem. Soc. 2018, 140, 37, 11560–11564) have shown that in their system one copy of the T7 terminator might not be sufficient to stall the T7 RNA polymerase.

Reviewer #2:

Remarks to the Author:

This manuscript describes the development of TRIDENT, a yeast-based in vivo mutagenesis platform. The authors initially developed TRIDENT by fusing the cytidine deaminase enzyme PmCDA1 to the T7 RNAP and found that the resulting system could introduce C to T mutations in a gene of interest that is downstream of a T7 promoter. To enhance the mutational diversity, the authors fused additional repair factors involved in processing of U:G intermediates in somatic hypermutation to the PmCDA1-T7 RNAP construct, and found that localizing MSH6 and APN2

allowed for high rates of G to A mutations as well as C to T. The authors then evolved an adenosine deaminase enzyme (TadA) to work in concert with T7 RNAP to further improve the system to induce mutations at A:T base pairs. Their final TRIDENT platform has both the PmCDA1-T7 RNAP and the TadA-T7 RNAP integrated in the yeast genome under inducible promoters, plus MSH6 and APN2 constructs that localize to the PmCDA1-T7 RNAP fusion protein, and the yeast UNG1 gene knocked out. The authors then utilize the TRIDENT system in 3 proof-of-principle experiments, demonstrating restoration of mCherry fluorescence via correction of a premature stop codon, engineering a 5 nm red-shifted mCherry variant, and engineering a DHFR variant with enhanced pyrimethamine resistance activity.

Overall, I think this is exciting work and a potentially very useful system. My main concern is that it seems like there is a lack of mutational diversity, and there are a couple spots in the manuscript that I think would benefit from additional clarifications.

1. There are several studies that suggest that increased UNG expression may actually help with mutational diversity of U:G intermediates (Nature Biotechnology, doi.org/10.1038/s41587-020-0609-x and doi.org/10.1038/s41587-020-0592-2, as well as biorxiv doi.org/10.1101/2020.07.21.213827). Have the authors looked at mutational diversity when UNG is not knocked out, or even recruited to the PmCDA1-T7 RNAP fusion protein as in the experiments from Figure 2? This may reduce off-target mutagenesis rates as well.
2. It would be helpful to quantify the mutational rates for each type of mutation of the final TRIDENT platform – similar to Figure 3C/D but showing every single type of mutation (i.e. C to T, C to A, C to G, etc., as in Figure 1E) on the same graph so that perspective users could see what they should expect to observe if they were to use the system.
3. The authors mention “Importantly, control of A/T and C/G bias could be implemented by independent expression of either deaminase.” Have they done this? It looks like mutations due to the cytidine deaminase (at C’s and G’s) dominate – at least this is my interpretation of Figure 3D.
4. Does the mutagenesis end abruptly at the end of the gene of interest? It isn’t clear to me where the end of the gene is in Figures 1D, 2B, and 3C.
5. Along these same lines, I think a more rigorous quantification of off-target mutagenesis is needed – if I am interpreting Figure S1 correctly, expression of the deaminase plus knockout of the UNG increases background mutagenesis rates from 10^{-8} to 10^{-4} ? This high background mutation rate can obviously greatly impact the success of directed evolution experiments and needs to be addressed. This same type of data need to be shown for the later TRIDENT systems.

Response to Reviewer Comments

We would like to thank the reviewers for their constructive comments and helpful suggestions for improving our manuscript. The comments were very useful in helping us to strengthen the manuscript and clarify the presentation of our work. In the sections below, we address each comment made by the reviewers.

Reviewer 1:

1. My main concern is about the off-target effects of TRIDENT. In the current study, the off-target mutation rate was measured via the fluctuation assay in a strain lacking the T7 promoter upstream of the URA3 gene. Have the authors looked at the mutation rates of other genomic regions other than URA3? On a related note, although it may be less of an issue for the purpose of directed evolution, have the authors examined if expressions of the TRIDENT system, especially in the presence of multiple engineered enzymes negatively impact the health status of the yeast cells?

We thank the reviewer for these thoughtful comments. As suggested, the mutation rate in yeast can vary widely depending on the genomic locus. To compare on/off-target mutation rates of our system, we controlled for this genomic variation by measuring the mutation rate at *URA3* in the presence or absence of a T7 promoter. We have not sequenced other sites in depth because the Illumina sequencing error rate is higher than the background mutation rate, and so would not inform off-target mutation rate. In addition, the core 18 base pair T7 promoter sequence is not present in the yeast genome, with no genomic site above 72% sequence similarity. This suggests that there would be minimal off-target effects driven by off-target T7 activity.

We conducted new experiments examining any fitness impact of the TRIDENT system. Supporting our hypothesis that there is limited off-target mutagenesis beyond what is measured at *URA3*, we found no statistically significant difference in growth rates between WT, YHM5, and YHM6 when induced with up to 100 nM β -estradiol. The results demonstrate the TRIDENT system has a minimal impact on cellular fitness. These data (Supplemental Table 3) have been added to the results in order to present the reader with additional information on the performance characteristics of the mutator strains.

2. I think for the purposes of continuous directed evolution, it would be very important to show multi-step fixation of mutations, and/or longer timescales of directed evolution of proteins with TRACE.

We thank the reviewer for this comment. We appreciate that multi-step fixation of mutations is a distinct experimental direction, but believe our work evolving DHFR highlights the ability of TRIDENT to introduce multiple mutations in a single gene. Therefore, we modified our analysis of DHFR evolution experiment to highlight the multiple mutations that have been previously identified as helping confer pyrimethamine resistance.

3. Page 3, line 78-81, although it is true that the TRACE system has a notable bias to C/G transition mutations, the authors' claim that it lacks tunability of overall mutation rate may not be accurate. It has been shown that the overall mutation rate can be tuned by using cytidine deaminase with different enzyme activity as well as using T7 RNA polymerase variants with different DNA processivity and elongation rate.

We thank the reviewer for drawing attention to these details of the TRACE system and have modified the discussion of TRACE to be more specific. In the discussion we changed “lacks tunability of overall mutation rate” to “lacks chemically inducible tunability of overall mutation rate” to clarify that TRIDENT allows for chemically inducible control over mutation rate.

4. *The authors should also acknowledge the work by Moore CL et al (J. Am. Chem. Soc. 2018, 140, 37, 11560–11564) and Álvarez B et al (bioRxiv, doi: <https://doi.org/10.1101/850974>).*

We thank the reviewer for this comment and have modified the introduction and discussion to include reference to the works by Moore *et al.* and Álvarez *et al.*

3. *Looking at the distribution of base substitution rates at the URA3 locus after the T7 promoter (e.g., Fig1D, Fig2B and Fig3C), there seems to be a change in base substitution rate as a function of the distance to the T7 promoter for certain types of substitution. Is it true? Have the authors quantified this?*

We thank the reviewer for noting a lack of discussion in the text regarding the distribution of base substitution rates as a function of distance from the T7 promoter. The C>T substitution rate is maintained up to 3 kb downstream of the T7 promoter as shown in Figure S5, and we added to the discussion to highlight this data. The G>A substitution rate appears to drop-off with distance from the T7 promoter, while our data indicate the T>C and A>G substitution rates have a more modest drop-off. We have modified the discussion to highlight the change in mutation rate as a function of distance from the T7 promoter for G>A, T>C and A>G substitutions.

4. *How long is the PfdHFR gene? I'd like to see some of the representative alignments of the Pacbio data on the evolved PfdHFR gene sequence. In particular, I'm interested to see if the T7 terminator at the end of the PfdHFR gene was able to stop the editing events since Moore CL et al (J. Am. Chem. Soc. 2018, 140, 37, 11560–11564) have shown that in their system one copy of the T7 terminator might not be sufficient to stall the T7 RNA polymerase.*

We thank the reviewer for this question and comment. The PfdHFR gene is 840 base pairs (280 amino acids). Pacbio sequencing data was obtained only between the promoter and terminator and did not provide information about potential mutagenesis beyond the T7 terminator. We did not analyze downstream mutations using PacBio sequencing as we expected the frequency of unenriched mutations would be similar to the inherent error rate of the Pacbio sequencing data. As noted, Moore CL et al. have shown that a single terminator is not sufficient to terminate mutagenesis. Similarly, Mairhofer J et al. (ACS Synth. Biol. 2015, 4, 265–27; <https://doi.org/10.1021/sb5000115>) quantified the efficiency of the native T7 terminator at approximately 80%. Based on these previously reported findings, we believe that a single T7 terminator likely prevents most, but not all, read-through and off-target mutagenesis in downstream chromosomal elements.

Based on the guidance from this comment, we amended the text to note that four or more copies of the T7 terminator may be used to eliminate TRIDENT read-through for experiments in crucial chromosomal regions.

Reviewer 2:

1. *There are several studies that suggest that increased UNG expression may actually help with mutational diversity of U:G intermediates (Nature Biotechnology, doi.org/10.1038/s41587-020-020-020-0, and doi.org/10.1038/s41587-020-020-020-0, as well as biorxiv doi.org/10.1101/2020.07.21). Have the authors looked at mutational diversity when UNG is not knocked out, or even recruited to the PmCDA1-T7 RNAP fusion protein as in the experiments from Figure 2? This may reduce off-target mutagenesis rates as well.*

We thank the reviewer for these insightful suggestions. In strains with PmCDA1 and UNG expression, while the background mutation rate was very low, we observed that the overall mutation rate was much lower as well. When UNG was targeted with SHL, we did not observe an improvement in TAA reversion rates, as shown in Supplementary Figure 7. In the noted publications, Cas9 is used to localize DNA-editing enzymes in mammalian cells - we suggest the difference in DNA localization mechanism (T7 RNAP vs. Cas9) and difference in host (yeast vs. mammalian) are responsible for our differing observations and have included this in the discussion.

2. It would be helpful to quantify the mutational rates for each type of mutation of the final TRIDENT platform – similar to Figure 3C/D but showing every single type of mutation (i.e. C to T, C to A, C to G, etc., as in Figure 1E) on the same graph so that perspective users could see what they should expect to observe if they were to use the system.

We thank the reviewer for this thoughtful suggestion. We have modified Figures 3C/D and added a new Supplementary Figure (Supplementary Figure 13).

3. The authors mention “Importantly, control of A/T and C/G bias could be implemented by independent expression of either deaminase.” Have they done this? It looks like mutations due to the cytidine deaminase (at C’s and G’s) dominate – at least this is my interpretation of Figure 3D.

We thank the reviewer for this comment. In our work, both DNA editing enzymes (yeTadA1.0 and PmCDA1) were controlled by the same β -estradiol inducible promoter. In the highlighted phrasing we intended to discuss the future directions of how TRIDENT might be used and modified. Such a future application might include placing yeTadA1.0 under control of a different inducible promoter so that PmCDA1 and yeTadA1.0 driven mutagenesis could be controlled by different chemical inducers. We have addressed this comment by clarifying this point in the discussion.

4. Does the mutagenesis end abruptly at the end of the gene of interest? It isn’t clear to me where the end of the gene is in Figures 1D, 2B, and 3C.

We thank the reviewer for this comment. Our initial optimizations of TRIDENT (Figures 1-3) did not include use of T7 terminators, and these were not included unless noted. In applications of TRIDENT to directed evolution challenges a single T7 terminator was used. Based on results from Moore *et al.* (J. Am. Chem. Soc. 2018, 140, 37, 11560–11564; <https://doi.org/10.1021/jacs.8b04001>) a single T7 terminator should prevent 80% of downstream transcription by T7 RNAP. We have addressed the termination of T7 RNAP mutagenesis by adding a discussion of the work by Moore *et al.* in which termination efficiency of T7 RNAP was investigated in detail.

5. Along these same lines, I think a more rigorous quantification of off-target mutagenesis is needed – if I am interpreting Figure S1 correctly, expression of the deaminase plus knockout of the UNG increases background mutagenesis rates from 10^{-8} to 10^{-4} ? This high background mutation rate can obviously greatly impact the success of directed evolution experiments and needs to be addressed. This same type of data need to be shown for the later TRIDENT systems.

We thank the reviewer for this comment. The interpretation of Figure S1 is correct; however, data shown in S1 was from the first generation of the system and formed part of the motivation for creating more optimized targeting systems, such as in Figure S3C. By using the SHL/SH3 domain system, the background *URA3* loss-of-function (LOF) rate was reduced to less than 10^{-5} . It is also important to note that the *URA3* LOF rate is much higher than the actual per-base mutation frequency, as many mutations within *URA3* can

contribute to the LOF phenotype. To address the concerns raised here, we have 1) clarified the discussion of Figure S1 data and 2) included new supplemental data (Supplementary Figure 13) detailing the off-target mutation rates of the most advanced TRIDENT strains, YHM5 and YHM6.

Reviewers' Comments:

Reviewer #1:

Remarks to the Author:

The authors have addressed my comments. I would suggest the authors deposit data into SRA over google drive. It would be useful to see the plasmids used in this study deposited in addgene as well.

Reviewer #2:

Remarks to the Author:

The authors have addressed all of my concerns and I believe the manuscript to be appropriate for publication in nature communications in its current form..

Response to Reviewer Comments

We would like to thank the reviewers for their constructive comments and helpful suggestions for improving our manuscript. The comments were very useful in helping us to strengthen the manuscript and clarify the presentation of our work. In the sections below, we address each comment made by the reviewers.

Reviewer 1:

1. The authors have addressed my comments. I would suggest the authors deposit data into SRA over google drive. It would be useful to see the plasmids used in this study deposited in addgene as well.

We thank the reviewer for their comment and time spent reviewing this paper. As suggested, we've deposited the data in SRA and deposited the most relevant plasmids into addgene.

Reviewer 2:

1. The authors have addressed all of my concerns and I believe the manuscript to be appropriate for publication in nature communications in its current form.

We thank the reviewer for their time.